# United States County-level COVID-19 Death Rates and Case Fatality Rates Vary by Region and Urban Status

**DOI:** 10.3390/healthcare8030330

**Published:** 2020-09-09

**Authors:** Rashid Ahmed, Mark Williamson, Muhammad Akhter Hamid, Naila Ashraf

**Affiliations:** 1College of Nursing, University of Manitoba, Winnipeg, MB R3T 2N2, Canada; Rashid.Ahmed@umanitoba.ca; 2School of Nursing and Professional Disciplines, University of North Dakota, Grand Forks, ND 58202, USA; 3Department of Pediatrics, University of Toronto, Toronto, ON M5S, Canada; Muhammad.Hamid@utoronto.ca; 4Southend Medical and Walk-in Clinic, Winnipeg, MB R2M 5G8, Canada; nailamalik1980@yahoo.com

**Keywords:** COVID-19, death rate, case fatality rate, county-level

## Abstract

COVID-19 is a global pandemic with uncertain death rates. We examined county-level population morality rates (per 100,000) and case fatality rates by US region and rural-urban classification, while controlling for demographic, socioeconomic, and hospital variables. We found that population mortality rates and case fatality rates were significantly different across region, rural-urban classification, and their interaction. All significant comparisons had *p* < 0.001. Northeast counties had the highest population mortality rates (27.4) but had similar case fatality rates (5.9%) compared to other regions except the Southeast, which had significantly lower rates (4.1%). Population mortality rates were highest in urban counties but conversely, case fatality rates were highest in rural counties. Death rates in the Northeast were driven by urban areas (e.g., small, East Coast states), while case fatality rates tended to be highest in the most rural counties for all regions, especially the Southwest. However, on further inspection, high case fatality rate percentages in the Southwest, as well as in overall US counties, were driven by a low case number. This makes it hard to distinguish genuinely higher mortality or an artifact of a small sample size. In summary, coronavirus deaths are not homogenous across the United States but instead vary by region and population and highlight the importance of fine-scale analysis.

## 1. Introduction

Severe acute respiratory syndrome coronavirus 2 (SARS-CoV-2) is the viral agent of Coronavirus Disease 2019 (COVID-19). Currently a global pandemic, it was first identified in Wuhan city, Hubei province, China and has since spread worldwide. By March 2020, the daily new cases were higher outside China than within [1]. By May 2020, over four million cases were reported across 187 countries [2]. The global case number has since passed 10 million, with almost 500,000 deaths. Furthermore, at the time of writing, the United States has the leading number of cases, with almost 2.9 million confirmed cases reported and over 130,000 deaths [3]. The Center for Disease Control and Prevention has reported an even higher number of cases and deaths of 3,047,671 and 132,056, respectively.

A variety of risk factors have been identified for COVID-19 infection and mortality. The most important are older age and underlying health conditions [4]. There are also indications that males face a higher mortality risk [5] and that members of ethnic minorities, especially Blacks, have higher infection rates [6,7].

The population mortality rate and the case-fatality rate are two important measures of disease impact. The population mortality rate is the number of deaths from a disease within a specified time adjusted by population size. Typically, this rate is multiplied by a set number, for example the mortality rate per 100,000. The case fatality rate is the proportion of cases that are fatal within a specified time. It may be underestimated because of a time-lag bias or overestimated because of case definition. As a disease emerges, the case fatality rate is overestimated at the beginning, but drops as more information is available. Overall, total cases are better than closed cases to express fatality rates [8].

Early cases from China (December 2019–April 2020) were analyzed to determine a case fatality rate of 5.65% for all China, with rates as high as 7.71% in Wuhan and as low as 0.86% in Hubei [9]. As of March 2020, the case fatality rate using total cases in China was 4% [8], while the May 2020 rate was 1.38% [10]. Outside China, the case fatality rate in Singapore, where contracting tracing and quarantine was implemented, was 0.30% in March 2020. A May 2020 report from France found a hospitalization rate of 3.6% and a population mortality rate of 0.7%, with mortality of up to 10.1% in the oldest age groups (>80 years old) [11]. A study using data from the Diamond Princess cruise ship provided one of the most robust case fatality rate estimates available because it was a contained population with few confounding factors and ample testing for the whole population. Results found a case fatality rate of 0.99%, though the older demographics do not represent the general population so overall rates are likely to be lower [12]. For the United States, the John Hopkins Center reported a gross case fatality rate of 4.8% in July 2020. A county-level study modeled the case fatality rate and estimated it to be much lower, at 1.3%, with county-level variation from 0.5% to 3.6% [13].

As the pandemic has continued, there has been an increased spread of COVID-19 infections into rural areas [14]. This is an issue because rural populations are more vulnerable due to comorbidities such as cancer, heart disease, diabetes, hypertension, and obesity [15,16]. Rural areas also have older age structures, which is expected to increase the expected death rate [17]. These vulnerabilities, combined with healthcare infrastructure shortfalls, make rural areas potentially high mortality areas [18]. A spatial study found hotspots in COVID-19 incidence and mortality rates per 100,000 in both urban and rural areas [19]. Another study found higher mortality rates per capita in rural, vulnerable counties compared to diverse, urban counties [20].

The United States is a large, heterogenous country with the most confirmed cases of COVID-19. The scope of this study encompasses counties in the United States across both region and rural-urban classifications. The United States can be broadly separated into five geographical regions, the Midwest, Northeast, Southeast, Southwest, and West. Each county can be further classified by a rural-urban code from 1 to 9. The most urban counties are classified as 1, the most rural as 9 (Appendix A).

It is important to understand the disease impact across different regions and populations. Our research will learn more about the impact of COVID-19 by modeling mortality rates using county-level data. Our main aim in this study is to determine the differences in COVID-19 death rates across regions and rural-urban classifications using case and death number data from the New York Times. We hypothesize that deaths per 100,000 will be higher on the East and West Coasts, as they are densely populated and have some of the earliest reported US cases. We also predict that case fatality rates will be no higher in urban areas compared to rural, because while rural areas may have less access to health facilities, urban areas are more prone to being overwhelmed by cases, leading to preventable deaths.

## 2. Materials and Methods 

### 2.1. Data Collection

Coronavirus case data at the county level were taken from a dataset created by the New York Times based on reports from state and local health agencies [21]. The most recent cases and deaths by county at the time of writing (23 June 2020) were used [21]. Rural-urban continuation codes from 2013 were obtained from the United States Department of Agriculture Economic Research Service [22]. Counties were classified into nine categories based on urban population and proximity to a metro areas (1 being the most metro, 9 being completely rural and not adjacent to a metro area). States were designated into one of five regions (Northeast, Midwest, West, Southwest, and Southeast). County-level demographic, social, and economic data were obtained from the 2018 American Community Survey [23]. Variables that may affect coronavirus infection and mortality rates were retained as confounding factors (Table 1). Sex ratio was the number of males per 100 females. Hospital bed data from Definitive Healthcare was obtained through ESRI and condensed to the county level [24]. Bed utilization was the proportion of total patient days over bed days available. All datasets were combined via FIPS county code using a custom Python dataset. Counties that had missing data were not used. New York City counties (Bronx, Kings, New York, Queens, and Richmond) were not individually available in the dataset but rather, were aggregated as New York City.

The response (dependent) variables used were coronavirus population mortality rates per 100,000 and coronavirus case fatality rates. The county population mortality rate per 100,000 was calculated by dividing the number of coronavirus deaths in the county by the county population (2010 US estimates) and multiplying it by 100,000. The county case fatality rate was calculated by dividing the number of coronavirus deaths in the county by the total number of county cases. After analysis, the proportions were multiplied by 100 to present them as percentages. Counties with zero cases were excluded from the analysis. The percentage of Whites, Blacks, and Hispanics, sex ratio, and age were all included as confounding factors because previous studies have shown differences in infection and mortality rates across ethnicity, sex, and age. Family size was included because larger families may be at greater risk of exposure, while the percentage of those that were uninsured or in poverty were included because those variables may impact on hospital access and care. The number of beds, bed utilization, and ventilator numbers were included because hospitals overwhelmed by COVID-19 hospitalizations may have had excess mortality rates.

### 2.2. Statistical Analysis

First, null models were created for each of the two response variables, population mortality rate per 100,000 and case fatality rate. Each model included the response variable as a function of region alone and rural-urban classification alone using a generalized linear model. A negative binomial distribution was used for the population mortality rate models over a Poisson distribution [25], because the rate data had a variance much higher than the mean, making the negative binomial a better fit [26,27]. A beta distribution was used for the case fatality rate models because the proportion data fit the range of the beta distribution, 0 < x < 1.

Next, demographic and socioeconomic variables were selected for inclusion as confounding factors in the population mortality rate models using stepwise model selection, a negative binomial distribution, and AICc score as the selection method. Then, for the case fatality rate models, because there is no appropriate model selection for a beta distribution, all confounding factors were included. Full models for each of the sets of models were run using the appropriate factors and distribution. Resulting least-square means for region and rural-urban classification were linked back to the data scale and plotted accordingly. Furthermore, population mortality rate and case fatality rate were each modeled as a function of the interaction of region and rural-urban classification by including the interaction term in the model formula, along with the single terms and all confounding factors. Finally, county-level population morality rates and case fatality rates were averaged by state and mapped by state and region.

## 3. Results

County-level population mortality rates per 100,000 and case fatality rates from COVID-19 were significantly different by region (Table 2a) and rural-urban classification (Table 2b) for both the null and full models (Appendix A). The full models had lower F-values and AICc scores than the null models, indicating that the confounding variables explained some but not all the variation. The interaction between region and rural-urban classification was also significant (Appendix A).

Population mortality rates were significantly different by region for both null (Figure 1a) and full (Figure 1b) models. The Northeast had a significantly higher rate than all other regions and the Midwest had a significantly higher rate than the Southwest and West. This pattern was expected given the prevalence of coronavirus infections in East Coast states. Case fatality rates were also significant by region for both null (Figure 1c) and full (Figure 1d) models. The Southeast had a significantly lower rate than other regions.

Population mortality rates were significantly different by rural-urban classification for both null (Figure 2a) and full (Figure 2b) models. Model-corrected mean morality rates per 100,000 were different across rural-urban continuum codes. Unsurprisingly, the most urbanized counties (code = 1) had a significantly higher rate than rural areas (codes = 7–9). Case fatality rates were also significant by rural-urban classification for both null (Figure 2c) and full (Figure 2d) models. Unlike population mortality rates, where the highest deaths per population were in the most urban counties, the most rural counties had the highest case fatality rates.

There was a significant interaction between region and rural-urban classification for both population morality rates and case fatality rates (see Appendix A for mean values and confidence intervals). For population mortality rates, the Southwest and Southeast did not have differences in rates by rural-urban classification. However, the Midwest and Northeast had higher death rates in the most urban areas and lower rates in the most rural areas. Finally, the West still had the lowest rates in the most rural areas, but had the highest rates in the most urban and the non-metro areas (code = 7) (Figure 3a). For case fatality rates, there were also differences by region. While most regions had the highest death percentages in the most rural areas, this difference was only significant in the Southwest and Midwest. The Southeast had relatively stable rates, while the Northeast, West, and Midwest followed a v-shaped pattern, where the lowest death percentages were in in counties with intermediate rural-urban classifications (Figure 3b).

Maps of population mortality rate and case fatality rate by state and region revealed that small states in the Northeast (New Jersey, Massachusetts, and Connecticut) had both the highest mortality rates and the highest case fatality rates. Western, Southwestern, and Midwestern states generally had very low population mortality rates, with the exceptions of Arizona and the eastern Midwest (Michigan, Indiana, Ohio), which had midrange rates (Figure 4a). Eastern Midwestern states also had higher case fatality rates, along with Louisiana. Utah and Alaska had extremely low case fatality rates (Figure 4b).

## 4. Discussion

### 4.1. Null versus Full Models

Based on the differences in the null and full models, demographic, socioeconomic, and hospital variables had an impact on death rates that justified their inclusion as confounding factors. However, there was an interesting shift between region and rural-urban classification. Population mortality rates retained the same pattern between the null and full model, indicating that accounting for other factors gave a similar result to region alone. In contrast, there was a significant difference in case fatality rates between the two models. Considering region alone, the Northeast had the highest percentage of deaths, followed by the Midwest. But adding in the other variables, all regions except the Southeast were comparable, while the Southeast was significantly lower. It was unclear why. Examining the means/medians for the confounding variables revealed that the Southeast had a higher percentage of residents in poverty and Black residents. Correcting for these variables—higher rates of each are expected to contribute to higher mortality—may have had some effect despite there being little difference between the null and full model. Other factors that could not be accounted for—such as comorbidities, environmental conditions, or policy decisions—may be stronger contributors to death rates.

For rural-urban classification, it was the case fatality rates that had the same pattern between the null and full models, while there was a significant difference between models for the population morality rates. The most urban counties still had the highest population mortality rates and the most rural counties had the lowest between models, but classifications between the two (codes = 2–8) were shifted up in the full model so that most were more comparable to the most urban counties. Accounting for average ventilator usage, percentage of Black residents, percentage of insured residents, and percentage of residents in poverty helped correct for the higher null rates found in the most urban areas.

### 4.2. Population Mortality Rates

As hypothesized, the region with the highest population mortality rate was the Northeast, though the same could not be said of the West. This is unsurprising, given that New York has the highest number of COVID-19 deaths, with 24,172 excess deaths over the baseline in New York City alone [28]. New Jersey, Massachusetts, Pennsylvania, and Connecticut also have fatality numbers among the top ten by state. Despite the high number of deaths, the case fatality rate was comparable to other regions except the Southeast, which had significantly lower death percentages. Looking at the interaction with rural-urban classification, the Southeast had similar death percentages across categories, while the rest of the regions had higher rates in more rural counties.

### 4.3. Case Fatality Rate 

Contrary to our predictions, the case fatality rate by rural-urban classification was the highest in the most rural counties rather than being the same across the rural-urban continuum. To explore why there were such high death percentages in completely rural counties, we examined case and death rates for completely rural Southwest counties in more detail (case = 9), as those counties had the highest case fatality rates. These included 37 counties in New Mexico, Oklahoma, and Texas. For those counties, the median case number was only 3.5, most counties had no deaths, and only five counties had a recorded death. Of those, the deaths per case were as high as 50% because two counties had two total cases and one death. Therefore, the high death percentage appeared skewed by low incidence. Looking at all counties, the death percentage was 0.00% at the 75% quartile and below. The high death percentages in rural areas appear to be driven by counties with few deaths and few cases.

### 4.4. State Maps

Mapping the population mortality rate and case fatality rate by state revealed more patterns. The higher population mortality rates in the Northeast were driven more by New Jersey, Massachusetts, and Connecticut than New York. Looking at New York by county revealed that while New York City had very high rates, many of the more rural counties had low or no mortality rates.

The Southwest had the lowest case fatality rates in the covariate-corrected model, a pattern that does not appear in the uncorrected map. Because the percentage of Blacks was a significant covariate, the Southwest had a higher percentage of Blacks than other regions and Blacks have been shown to be more susceptible to COVID-19 infection [6,29]. Correcting for the percentage of Black residents may be a reason that the model-corrected case fatality rates were so low in the Southwest. As mentioned before however, there still appears to be unexamined factors that are contributing to low regional death rates.

### 4.5. Other Factors

Given the regional distribution of mortality rates, there may be significant factors at work that were not available for this analysis. Two possible environmental factors are air pollution and temperature. Fine particulate matter was positively associated with COVID-19 death rates in a recent US study [30]. While that association corroborates with our findings that more urban counties have higher population mortality rates, it does not account for the higher case fatality rates found in more rural counties. A higher daily temperature range in January–February was positively associated with COVID-19 death rates in a study on Wuhan, China [31], while an average monthly high temperature in April was negatively associated with COVID-19 death rates in a study of Western counties [32]. No pattern seemed evident on the state maps regarding temperature, either positive or negative. The effect of temperature is also likely to change across seasons. Policy decisions are also expected to influence mortality rates. Some possible factors that were not included are: lockdown status, nursing home policy, testing regiment, and mask mandates. Finally, there are immunological characteristics that may lower certain populations’ susceptibility to COVID-19. Studies of the current pandemic and previous ones suggests that this is likely [33,34].

## 5. Conclusions

This study had several limitations. First, the data was aggregated at the county level, so individual metrics were not available. We do not know if the population that made up the cases and deaths have the same demographics as the county-level population. Second, not all possible confounders of importance—such as comorbidities, environmental differences, policy differences, or immunological factors—were available. For example, counties that have residents with more health issues will certainly have higher mortality rates. Third, it is known that there is a significant asymptomatic rate of infection. The true death rate is impossible to measure from case fatality rate alone, as only reported cases are counted. Finally, there were many counties that had few cases or deaths, mostly found in rural areas of the Midwest and Southeast. This led to some counties having a very high or low case fatality rate that was more a function of small sample size than actual factors that made it more or less likely to die from a COVID-19 infection.

More work remains to be done to explain regional and rural–urban differences. As SARS-CoV-2 continues to spread through the most rural areas, a more accurate picture of the mortality rate and case fatality rate will emerge. This will lead to a better understanding of how and where to allocate resources to help vulnerable populations.

## Figures and Tables

**Figure 1 healthcare-08-00330-f001:**
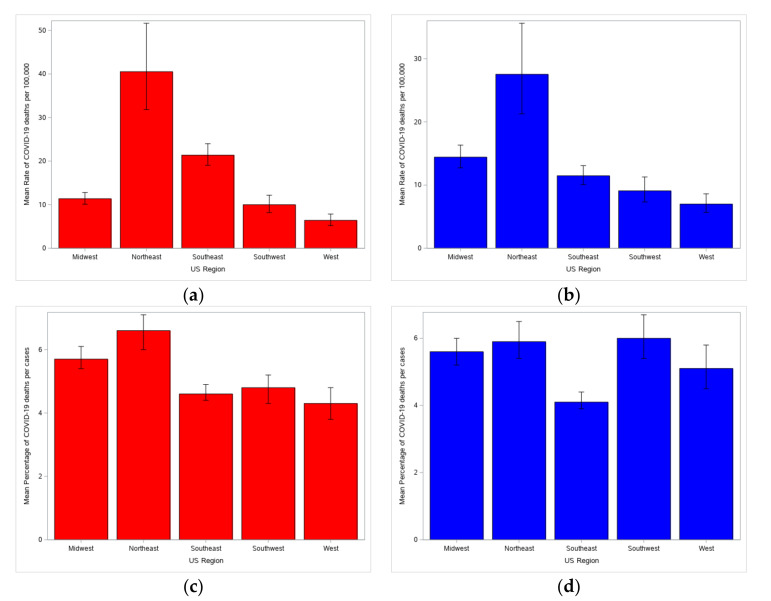
Differences in rate by region. (**a**) Mean death rate per 100,000 for region-only model. (**b**) Mean death rate per 100,000 for full model (included confounding factors). (**c**) Mean percentage of deaths per cases for region-only model. (**d**) Mean percentage of deaths per cases for full model (included confounding factors).

**Figure 2 healthcare-08-00330-f002:**
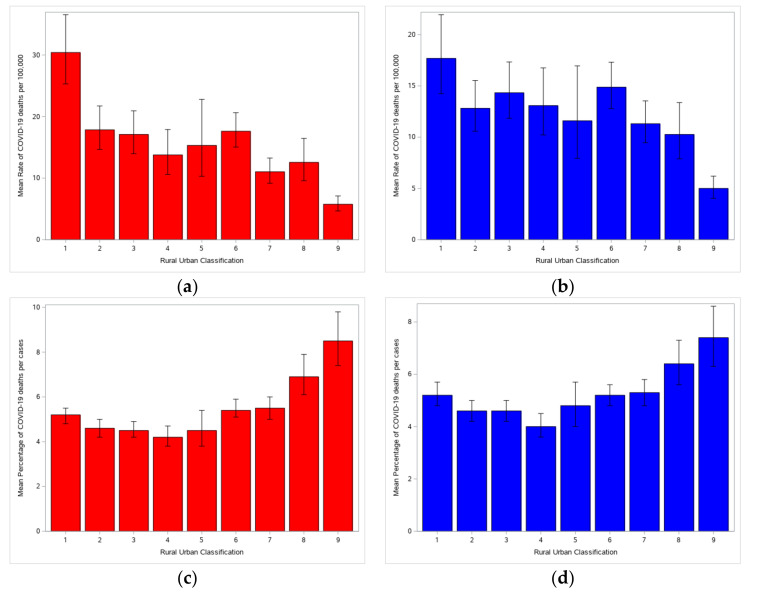
Differences in rate by rural-urban designation (1 = most urban, 9 = most rural). (**a**) Mean death rate per 100,000 for the rural-urban-only model. (**b**) Mean death rate per 100,000 for full model (including confounding factors). (**c**) Mean percentage of deaths per cases for rural-urban-only model. (**d**) Mean percentage of deaths per cases for full model (including confounding factors).

**Figure 3 healthcare-08-00330-f003:**
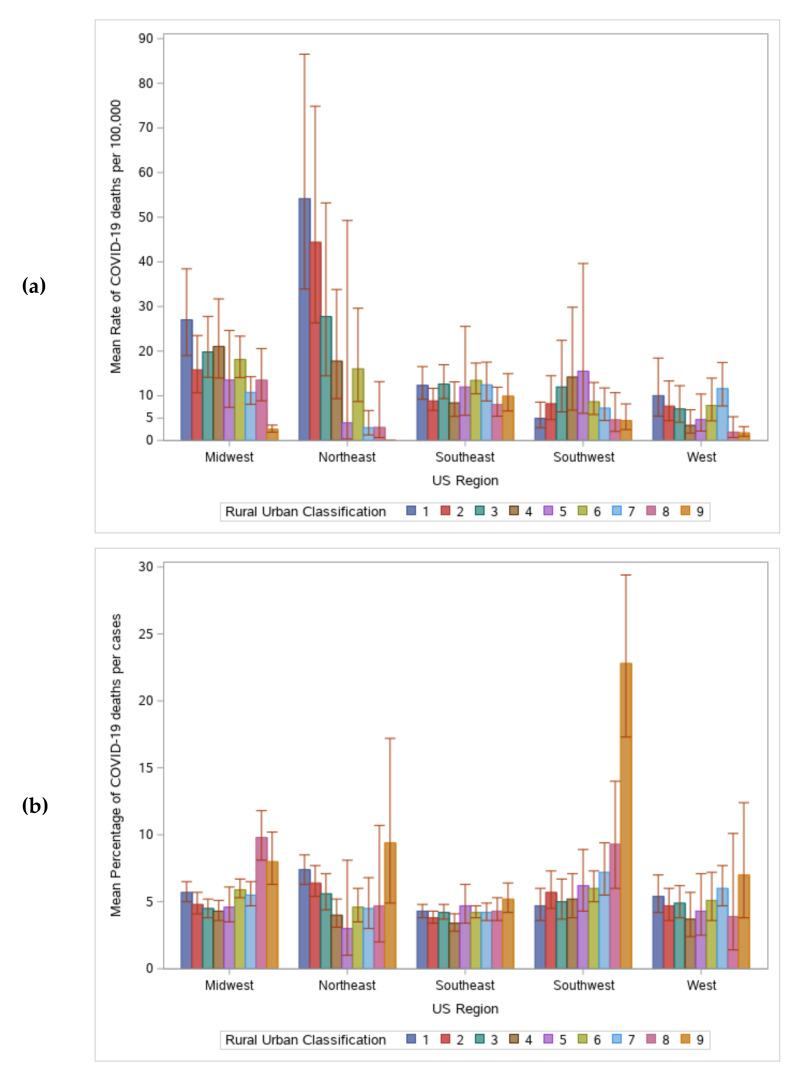
Differences in rates by the interaction between region and rural-urban designation (1 = most urban, 9 = most rural). Models included confounding factors. (**a**) Mean death rate per 100,000. (**b**) Mean percentage of deaths per cases.

**Figure 4 healthcare-08-00330-f004:**
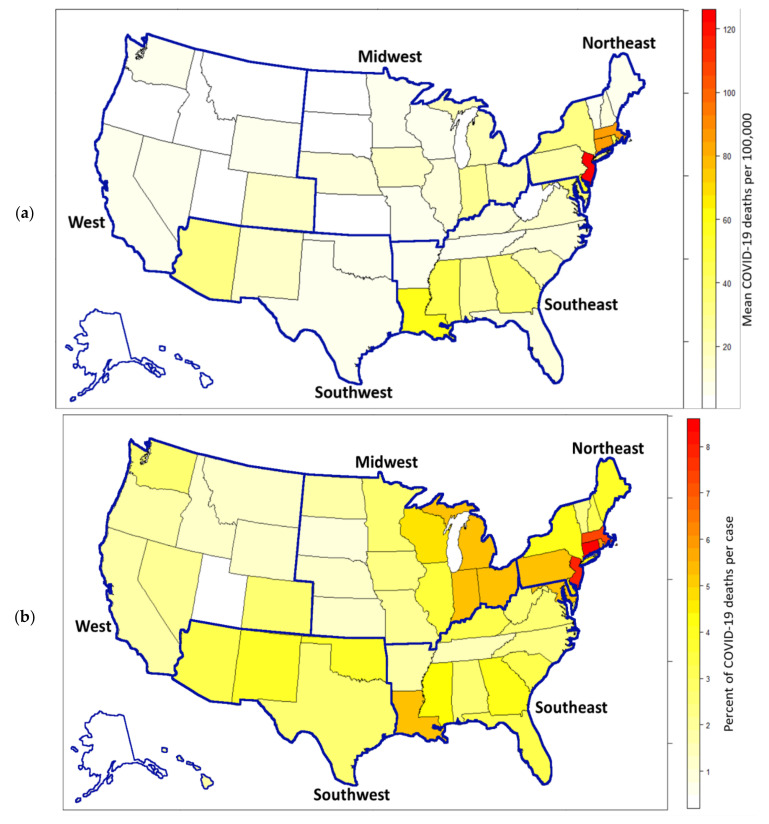
State and Region maps. (**a**) Mean COVID-19 mortality rate per 100,000. (**b**) Mean Percentage of COVID-19 deaths per case. Values were calculated by averaging county-level rates across each state. Alaska and Hawaii are part of in the Western region and were moved and scaled to fit.

**Table 1 healthcare-08-00330-t001:** Factors from the American Community Survey and Definitive Healthcare Hospital Beds data.

Variable	Total US		Midwest		Northeast		Southeast		Southwest		West	
	Mean	Median	Mean	Median	Mean	Median	Mean	Median	Mean	Median	Mean	Median
**Demographic**												
% White	83.1	89.7	91.4	94.7	85.3	91.2	74.8	78.8	81.3	83.5	83.9	89.8
% Black	9.1	2.3	2.6	0.9	7.1	3.3	20.3	14.2	5.2	2.9	1.3	0.7
% Hispanic	9.3	4.1	4.6	2.8	6.8	3.8	5.0	3.2	30.6	23.5	14.4	9.4
Sex ratio	100.9	98.5	100.9	99.7	99.1	97.1	98.8	96.2	103.8	99.4	104.8	101.6
Age	41.3	41.2	42.0	41.9	42.5	42.5	41.1	41.0	39.4	39.0	41.1	40.2
**Socioeconomic**												
Family size	3.1	3.0	3.0	2.9	3.0	3.0	3.1	3.1	3.3	3.2	3.2	3.1
Income	51,557	49,886	53,508	52,558	63,285	59,114	45,620	42,621	49,727	48,331	56,827	53,311
% insured	89.9	90.8	92.2	93.3	94.0	94.5	89.1	89.2	83.9	84.3	89.3	90.5
% in poverty	11.2	10.3	8.8	8.2	8.2	8.0	14.5	13.7	12.5	11.8	9.5	9.0
**Hospital**												
Bed number	305.2	42.0	205.7	25.0	813.4	237.0	251.9	55.0	314.9	25.0	389.5	30.0
Bed utilization	0.3	0.3	0.3	0.3	0.5	0.5	0.3	0.3	0.2	0.2	0.3	0.3
Ventilator number	2.0	2.0	1.5	1.0	3.7	3.0	2.3	2.0	1.4	1.0	1.7	1.5

**Table 2 healthcare-08-00330-t002:** Rate estimates from final models across region and rural-urban code.

**a. Model-corrected mean mortality rate per 100,000 mean and case fatality rate percentage (CFR %) by US region. Confidence Intervals (CI) are 95%.**						
**Predictor Variable**	**Mean Mortality Rate per 100,000**			**CFR %**		
**US Region**	**Estimate**	**Lower CI**	**Upper CI**	**Estimate**	**Lower CI**	**Upper CI**
**Midwest**	14.4	12.7	16.3	5.6%	5.2	6.0
**Northeast**	27.5	21.3	35.6	5.9%	5.4	6.5
**Southeast**	11.5	10.1	13.1	4.1%	3.9	4.1
**Southwest**	9.1	7.3	11.3	6.0%	5.4	6.7
**West**	7.0	5.7	8.6	5.1%	4.5	5.8
**b. Model-corrected mean mortality rate per 100,000 mean and case fatality rate percentage (CFR %) by Rural-Urban Classification. Confidence Intervals (CI) are 95%.**						
**Predictor Variable**	**Mean Mortality Rate per 100,000**			**CFR %**		
**Rural-Urban Code**	**Estimate**	**Lower CI**	**Upper CI**	**Estimate**	**Lower CI**	**Upper CI**
**1**	17.7	14.2	21.9	5.2%	4.8	5.7
**2**	12.8	10.6	15.5	4.6%	4.2	5.0
**3**	14.3	11.8	17.3	4.6%	4.2	5.0
**4**	13.1	10.2	16.7	4.0%	3.6	4.5
**5**	11.6	7.9	17.0	4.8%	4.0	5.7
**6**	14.9	12.8	17.3	5.2%	4.8	5.6
**7**	11.3	9.5	13.5	5.3%	4.8	5.8
**8**	10.3	7.9	13.4	6.4%	5.6	7.3
**9**	5.0	4.0	6.2	7.4%	6.3	8.76).

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
