# Peer review of "United States County-level COVID-19 Death Rates and Case Fatality Rates Vary by Region and Urban Status"

_healthcare, 2020, doi:10.3390/healthcare8030330_

Round 1

Reviewer 1 Report

This paper had the potential of being an interesting case study on regionality of COVID-19 in the USA, and especially urban/rural differences, however the study is lacking in many areas, including basic description on the variables and how they are generated, and specific values and tests seem to be missing in the results. The introduction is lacking in background information on regionality of infectious diseases and COVID-19, and the importance of the study needs to be emphasized.

Introduction

I believe the following statement “…patients that were Black, Asian, or another ethnic minority were overrepresented…”, should be “patients that were Black, Asian, or another ethnic minorities were overrepresented..” In the study that you are referring, is occupation a factor, as many of the ethnic minorities could be “frontline” workers. When discussing minority groups, there should also be a discussion of urban risk for COVID.

Why have you included China id there is so much error in the estimate of mortality and case-fatality rates, if censoring and ascertainment biases only account for 1.38%, does that mean that other than that China in actuality has a low rate relative to the rest of world? What else could account for the very low rates, perhaps China is best left out of the discussion because of the obvious limitations with regards to the accuracy of the data, and that it should be mentioned as not being appropriate for comparison.

For line 65, please describe what the “oldest age groups” are. What ages are we referring to here, 65 years and older?

What does the Diamond Princess cruise ship provide the most robust CFR estimate? Is this because it was an outbreak among a contained population? Please describe.

I am confused by the following statement on lines 68-70: “Within the United States, there are contrasting fatality reports. The John Hopkins Center reports a 4.8% CFR for the US. However an estimate for the US infection fatality rate was 1.3%, with county-level variation from 0.5% to 3.6%.” Isn’t an infection fatality suppose to be lower because it would encompass all cases (some sort of estimate of asymptomatics)?

There is no attempt made to discuss comparison of urban-rural studies of COVID in other nations or in the USA, for that matter. There needs to be a summary of urban/rural differences in previous studies on COVID. There is at least one study: Paul, R., Arif, A. A., Adeyemi, O., Ghosh, S., & Han, D. (2020). Progression of COVID‐19 From Urban to Rural Areas in the United States: A Spatiotemporal Analysis of Prevalence Rates. The Journal of Rural Health.

In general, there need to be way more info on the “demographic and socioeconomic factors affect infection and mortality rates, so such factors need to be accounted for..” What demographic and socioeconomic factors?Please specify. There needs to be way more clarity as to the factors that have been studied, which are suspected to contribute to heterogeneity in cases, and mortality and severity (case-fatality).

Why are “hospital variables such as bed and ventilator number are also inspected suspected to have an impact on mortality rate? Please explain and support with references.

Please provide a bit more clarity as the areas of study, that is it done by regions (describe them briefly, as you have done in the methods).

Can you provide a link the to the data since this is publicly accessible data?

What are the “other important factors?” You can not leave this up to the reader to guess, it is important you provide clear objectives, what factors are you including and why? There need to way more attention paid to this rather than the comparison of CFR to other countries.

There should be some sort of summary of population density for the different regions and by urban and rural, if this is factor you are including in your study. Please include in a table. I am assuming you mean East and West Coasts on line 83.

Materials & Methods

A map showing the counties included in the study would add perspective.

Please also show the regions and, if possible where there are urban communities and rural communities.

Again, what are the variable that are in line 99: “Variables that may affect Coronavirus infection and death rate were retained.”

The following sentence is somewhat awkward: “Two response variables were created using coronavirus case and death data.” I believe it could just be “The response (dependent) variables are coronavirus mortality rates per 100000 and deaths per cases. Can you please explain why you have not use the case fatality rate (CFR), the deaths per cases multiplied by 100 (usually given as a percent)?

I am confused by table 1, as you state that you are analyzing the co-variance (correlation), of the variables, so my understanding is that these are your variables of interest, however, you havedescribed these as “confounding variables” in the table caption. I don’t think this is the best term to use here, I understand that the variables could be considered confounders in null models, but they may be important variables in the full model. Also please describe how these variables are constructed, what is used to determine poverty line for example? Is bed utilization a percent of the number of beds?

Can you please provide references to support why you have used a negative binomial?

When you refer to “For the second set” do you mean the analyses on deaths per cases? Please specify.

Please describe how you tested for interaction.

Results

There should be some descriptive summary of the COVID rates especially by region. The boxplots don’t give the exact values.

For Figure 1 there is mention of significant differences, but no indication of a test or p-values are given. Can not a similar tests be done for Figures 2 and 3?

Also, the rural communities with low cases and/or deaths should not be included in the study. I can’t tell which counties these are; some description is provided in the discussion

Discussion

Rural higher case fatality, the analysis of CFR in rural areas should be removed because of the low number of cases and deaths

In the discussion you state: “Southeast was significantly lower. It was unclear why.”, can you offer any speculation as to why the Southeast had the lowest case-fatality? Couldn’t age distribution be playing a role?

I like the limitations section, I would replace” individual metrics” this is confusing, it sounds like you are referring to individual people, please rephrase to “data for each county,” or something similar.

For explanation for urban-rural you can refer to the literature on past pandemics, that will offer insight, such as previous exposure to infectious diseases, isolation, and access to health care. There is quite a bit of literature on this with regard to 1918 influenza.

See: Mathews, J. D., McBryde, E. S., McVernon, J., Pallaghy, P. K., & McCaw, J. M. (2010). Prior immunity helps to explain wave-like behaviour of pandemic influenza in 1918-9. BMC infectious diseases, 10(1), 1. And Worobey, M., Han, G. Z., & Rambaut, A. (2014). Genesis and pathogenesis of the 1918 pandemic H1N1 influenza A virus. Proceedings of the National Academy of Sciences, 111(22), 8107-8112.

Minor Grammatical issues:

There are too many sentences that begin with “however”. And there are many sentences that are too short and could be elaborated.

Author Response

Reviewer 1 :

I believe the following statement “…patients that were Black, Asian, or another ethnic minority were overrepresented…”, should be “patients that were Black, Asian, or another ethnic minorities were overrepresented..” In the study that you are referring, is occupation a factor, as many of the ethnic minorities could be “frontline” workers. When discussing minority groups, there should also be a discussion of urban risk for COVID.

  • Sentence was cut in revision
  • Referenced study (Raisi-Estabragh et al., 2020) did not include occupational factors, so no information about frontline workers is known
  • Added a section on rural/urban risk [lines 64-71]

Why have you included China id there is so much error in the estimate of mortality and case-fatality rates, if censoring and ascertainment biases only account for 1.38%, does that mean that other than that China in actuality has a low rate relative to the rest of world? What else could account for the very low rates, perhaps China is best left out of the discussion because of the obvious limitations with regards to the accuracy of the data, and that it should be mentioned as not being appropriate for comparison.

  • Revised section to only include relevant points on China’s case fatality rate
  • Takeaway point of including China was that case fatality rates dropped as time went one

For line 65, please describe what the “oldest age groups” are. What ages are we referring to here, 65 years and older?

  • Added description (> 80 years old) [now line 57]

What does the Diamond Princess cruise ship provide the most robust CFR estimate? Is this because it was an outbreak among a contained population? Please describe.

  • Added more detail [lines 57-59]

I am confused by the following statement on lines 68-70: “Within the United States, there are contrasting fatality reports. The John Hopkins Center reports a 4.8% CFR for the US. However an estimate for the US infection fatality rate was 1.3%, with county-level variation from 0.5% to 3.6%.” Isn’t an infection fatality suppose to be lower because it would encompass all cases (some sort of estimate of asymptomatics)?

  • I revised this section. The John Hopkins Center had a gross national rate, while the study estimated a rate by county.  It should be clearer now [lines 61-63]

There is no attempt made to discuss comparison of urban-rural studies of COVID in other nations or in the USA, for that matter. There needs to be a summary of urban/rural differences in previous studies on COVID. There is at least one study: Paul, R., Arif, A. A., Adeyemi, O., Ghosh, S., & Han, D. (2020). Progression of COVID‐19 From Urban to Rural Areas in the United States: A Spatiotemporal Analysis of Prevalence Rates. The Journal of Rural Health.

  • Added a section on rural/urban studies [lines 64-71]

In general, there need to be way more info on the “demographic and socioeconomic factors affect infection and mortality rates, so such factors need to be accounted for..” What demographic and socioeconomic factors?  Please specify. There needs to be way more clarity as to the factors that have been studied, which are suspected to contribute to heterogeneity in cases, and mortality and severity (case-fatality).

  • Deleted this section, added earlier section on risk factors [lines 39-42]

Why are “hospital variables such as bed and ventilator number are also inspected suspected to have an impact on mortality rate? Please explain and support with references.

  • Deleted this section.

Please provide a bit more clarity as the areas of study, that is it done by regions (describe them briefly, as you have done in the methods).

  • Added a paragraph on area of study [lines 72-77]

Can you provide a link the to the data since this is publicly accessible data?

  • Yes, included it as a reference (#21)

What are the “other important factors?” You can not leave this up to the reader to guess, it is important you provide clear objectives, what factors are you including and why? There need to way more attention paid to this rather than the comparison of CFR to other countries.

  • Deleted this section. Explained across a few paragraphs the reasoning for what factors were chosen and why [lines 95-115]

There should be some sort of summary of population density for the different regions and by urban and rural, if this is factor you are including in your study. Please include in a table. I am assuming you mean East and West Coasts on line 83.

  • I am not including population density as a factor, but do use county population to calculate population mortality rate per 100,000 [lines 104-106]
  • Yes, meant East and West Coasts, corrected it

Materials & Methods

A map showing the counties included in the study would add perspective.

  • Created map and included in supplemental information (Supplemental Figure 1a)

Please also show the regions and, if possible where there are urban communities and rural communities.

  • Created map and included in supplemental information (Supplemental Figure 1b). Regions available in newly-added Figure 4

Again, what are the variable that are in line 99: “Variables that may affect Coronavirus infection and death rate were retained.”

  • Updated this section and referenced Table 1 as the identities of the variables

The following sentence is somewhat awkward: “Two response variables were created using coronavirus case and death data.” I believe it could just be “The response (dependent) variables are coronavirus mortality rates per 100000 and deaths per cases. Can you please explain why you have not use the case fatality rate (CFR), the deaths per cases multiplied by 100 (usually given as a percent)?

  • Updated the sentence for clarity
  • Also, results now have the CFR as a percentage rather than a proportion [lines 103-108]
  • Figures 1c, 1d, 2c, 2d, and 3b all are now in percentage

I am confused by table 1, as you state that you are analyzing the co-variance (correlation), of the variables, so my understanding is that these are your variables of interest, however, you havedescribed these as “confounding variables” in the table caption. I don’t think this is the best term to use here, I understand that the variables could be considered confounders in null models, but they may be important variables in the full model. Also please describe how these variables are constructed, what is used to determine poverty line for example? Is bed utilization a percent of the number of beds?

  • I used covariates imprecisely. I wanted them as confounding variables (not interested in their effect, just wanted to control for their effect).  I have updated the wording to use ‘confounding’ in place of ‘covariate’
  • Variables were taken from the 2018 American Community Survey [lines 94-95] which were county level data
  • I included variable explanations for Sex Ratio [line 96] and Bed Utilization [lines 97-98], as they were not clear

Can you please provide references to support why you have used a negative binomial?

  • Added references (#26, #27)

When you refer to “For the second set” do you mean the analyses on deaths per cases? Please specify.

  • Updated section for clarity

Please describe how you tested for interaction.

  • Added more clarity on the model formula [lines 132-136]

Results

There should be some descriptive summary of the COVID rates especially by region. The boxplots don’t give the exact values.

  • Results now have either a descriptive summary [lines 146-155, 163-174] or are included in the supplemental information (Supplemental Tables 1 and 2)

For Figure 1 there is mention of significant differences, but no indication of a test or p-values are given. Can not a similar tests be done for Figures 2 and 3?

  • Figures 1, 2, and 3 are all the results from the models outlines in Table 2
  • Each model has the F-test statistic and p-value within the table

Also, the rural communities with low cases and/or deaths should not be included in the study. I can’t tell which counties these are; some description is provided in the discussion

  • Counties with no cases were excluded from the study
  • For counties with low cases and/or deaths, I believed they were important to retain in the study because our aim was to estimate mortality with the available data, even if that estimate is an over or under-estimate
  • I have made the caveats of these counties’ results clear, so the reader understands the caveats and can make their own decisions. I also added more information about rural counties with low cases/deaths [lines 290-293]

Discussion

Rural higher case fatality, the analysis of CFR in rural areas should be removed because of the low number of cases and deaths

  • See above

In the discussion you state: “Southeast was significantly lower. It was unclear why.”, can you offer any speculation as to why the Southeast had the lowest case-fatality? Couldn’t age distribution be playing a role?

  • Explored the question of the Southeast in more detail [lines 215-221, 260-266]
  • As seen in the update to Table 1, the median age is little different from other regions and the US overall.

I like the limitations section, I would replace” individual metrics” this is confusing, it sounds like you are referring to individual people, please rephrase to “data for each county,” or something similar.

  • I meant to talk about individual people (how the population of individuals that made up the cases and deaths may not match the demographics of the county). I added a sentence to explain [lines 284-285]

For explanation for urban-rural you can refer to the literature on past pandemics, that will offer insight, such as previous exposure to infectious diseases, isolation, and access to health care. There is quite a bit of literature on this with regard to 1918 influenza.

  • I offered a brief point on immunology, referencing both the paper below and a current COVID-19 paper (#33, #34)

See: Mathews, J. D., McBryde, E. S., McVernon, J., Pallaghy, P. K., & McCaw, J. M. (2010). Prior immunity helps to explain wave-like behaviour of pandemic influenza in 1918-9. BMC infectious diseases, 10(1), 1. And Worobey, M., Han, G. Z., & Rambaut, A. (2014). Genesis and pathogenesis of the 1918 pandemic H1N1 influenza A virus. Proceedings of the National Academy of Sciences, 111(22), 8107-8112.

Minor Grammatical issues:

There are too many sentences that begin with “however”. And there are many sentences that are too short and could be elaborated.

  • Removed excess ‘however’ statements
  • Edited document for short sentences

Reviewer 2 Report

Dear Authors, thank you very much for your manuscript. It is a good draft but very hard to judge the merits of the research. The analysis is not presented in an adequate manner in order to conclude whether the results are calculated correctly. Introduction should be revised because it does not lead the reader but stems into different directions. You should consider to revise and keep only you deem important and only that to state the start point and where your research contributes. The analysis description has many grey spots, has to be explained better. Conclusions are rather short and leaves a lot for the reader to figure out alone. Abstract has to be revised completely without any values and more substance that summarizes the paper.

Author Response

Reviewer 2 :

The analysis is not presented in an adequate manner in order to conclude whether the results are calculated correctly.

  • Increased clarity of methods
    • Updated Table 1 to include mean/median information on the factors across region
    • Added subjection headers
    • Revised method test to more clearly explain variables used, how the response variables were calculated, and standardized use of the terms ‘population mortality rate’, and ‘case fatality rate’
  • Presented results more clearly
    • Changed case fatality rates form proportions to percentages
    • Included numerical values for each of the tests

Introduction should be revised because it does not lead the reader but stems into different directions. You should consider to revise and keep only you deem important and only that to state the start point and where your research contributes.

  • Modified section on infection/mortality rates to focus on general risk factors, not just ethnic factors
  • Made population mortality rate and case fatality rate section clearer
  • Excluded irrelevant information on case fatality rates across countries and added relevant information about rural/urban differences
  • Added scope of study section and cleaned up the aims section

The analysis description has many grey spots, has to be explained better.

  • Added subsections for the methods to make it clearer
  • Rewrote parts of the Statistical Analysis section and added justification for the negative binomial

Conclusions are rather short and leaves a lot for the reader to figure out alone.

  • Added a sentence of explanation to each conclusion point
  • Added another conclusion point

Abstract has to be revised completely without any values and more substance that summarizes the paper.

  • I looked at other abstracts published in Healthcare
    • Based on what I saw, I dropped most of the values and just kept the rates or percentages on key results
  • Updated information on results

Reviewer 3 Report

This paper examined the county-level death rate and CFR by US region and rural-urban classification while controlling for demographic, socioeconomic, and hospital variables. It found that coronavirus deaths are not homogenous across the United States instead vary by region and population. The design is appropriate, and the results are clearly presented. I only have a few comments on it.

  1. Table 1 should be reorganized with a statistical summary of these covariates.
  2. It is better to add US maps showing confirmed cases and deaths for each region.
  3. Considering the regional distribution of COVID-19 death rate and CFR shown in Figure 2, do environmental factors, especially air pollution, have an impact on the death rate or CFR?

Wu, X., Nethery, R. C., Sabath, B. M., Braun, D., & Dominici, F. (2020). Exposure to air pollution and COVID-19 mortality in the United States. medRxiv. (DOI: https://doi.org/10.1101/2020.04.05.20054502)

Ran, J., Zhao, S., Han, L., Qiu, Y., Cao, P., Yang, Z., ... & He, D. (2020). Effects of Particulate Matter Exposure on the Transmissibility and Case Fatality Rate of COVID-19: A Nationwide Ecological Study in China. Journal of Travel Medicine. (DOI: https://doi.org/10.1093/jtm/taaa133)

Author Response

Reviewer 3:

Table 1 should be reorganized with a statistical summary of these covariates.

  • Table 1 was updated with means and medians for each variable across region and for the US overall

It is better to add US maps showing confirmed cases and deaths for each region.

  • Added Figure 4, which showed population mortality rates per 100,000 and case fatality rates for each state
  • Regions were a

Considering the regional distribution of COVID-19 death rate and CFR shown in Figure 2, do environmental factors, especially air pollution, have an impact on the death rate or CFR?

  • Added a section in the discussion (4.5 Other Factors) that discussed environmental factors and other possible factors that may affect death rate or CFR [lines 267-281]

Round 2

Reviewer 2 Report

Dear Authors,

thank you very much for your updated manuscript.

I would recommend to move Table 1 to appendix, and instead of F-values present AICc results. And probably it would be more beneficial for the readers to have a table with the CI values that are very densely populating the text. 

I would recommend a thorough inspection and editing of the manuscript, eg. Figure 3 label says bar charts of null and full models, and it is not clear what is in graph actually but not both, I would assume.

Also the current format has to be checked with the journal together, some figures are cut by a page break. Please make sure all the requirements are met.

Author Response

Reviewer 2 additional Feedback:

I would recommend to move Table 1 to appendix, and instead of F-values present AICc results. And probably it would be more beneficial for the readers to have a table with the CI values that are very densely populating the text.

  • I believe you meant Table 2, as Table 1 does not have F-values
  • Moved Table 2 to supplemental text (kept some text on info) and added both AICc scores and Pearson Chi-squared/DF values
  • Created table for Mortality rates and Case fatality rates estimates (Table 2)

I would recommend a thorough inspection and editing of the manuscript, eg. Figure 3 label says bar charts of null and full models, and it is not clear what is in graph actually but not both, I would assume.

  • Inspected and edited abstract, introduction, methods, results, graphs and tables, discussion, conclusion, references, and supplemental information
  • Labels for Figures 1-3 were edited for clarity

Also the current format has to be checked with the journal together, some figures are cut by a page break. Please make sure all the requirements are met.

  • Checked the revised manuscript against the Healthcare template document
  • Checked the revised manuscript against the Healthcare Instructions for Authors
  • Ensured figures were not cut off